# Learned-Database Systems Security

**Roei Schuster**                                                                 *rschuster@gmail.com*
*Context AI*

**Jin Peng Zhou**                                                                 *jz563@cornell.edu*
*Cornell University Department of Computer Science*

**Thorsten Eisenhofer**                                          *thorsten.eisenhofer@tu-berlin.de*
*BIFOLD & TU Berlin*

**Paul Grubbs**                                                                  *paulgrub@umich.edu*
*University of Michigan*

**Nicolas Papernot**                                          *nicolas.papernot@utoronto.ca*
*University of Toronto & Vector Institute*

**Reviewed on OpenReview:** *https://openreview.net/forum?id=XNVBSbtcKB*

## Abstract

A learned database system uses machine learning (ML) internally to improve performance. We can expect such systems to be vulnerable to some adversarial-ML attacks. Often, the learned component is shared between mutually-distrusting users or processes, much like microarchitectural resources such as caches, potentially giving rise to highly-realistic attacker models. However, compared to attacks on other ML-based systems, attackers face a level of indirection as they cannot interact directly with the learned model. Additionally, the difference between the attack surface of learned and non-learned versions of the same system is often subtle. These factors obfuscate the de-facto risks that the incorporation of ML carries. We analyze the root causes of potentially-increased attack surface in learned database systems and develop a framework for identifying vulnerabilities that stem from the use of ML. We apply our framework to a broad set of learned components currently being explored in the database community. To empirically validate the vulnerabilities surfaced by our framework, we choose 3 of them and implement and evaluate exploits against these. We show that the use of ML cause leakage of past queries in a database, enable a poisoning attack that causes exponential memory blowup in an index structure and crashes it in seconds, and enable index users to snoop on each others' key distributions by timing queries over their own keys. We find that adversarial ML is an universal threat against learned components in database systems, point to open research gaps in our understanding of learned-systems security, and conclude by discussing mitigations, while noting that data leakage is inherent in systems whose learned component is shared between multiple parties.

## 1 Introduction

*Learned-database systems* incorporate ML models internally to improve performance. These models adapt the system's inner workings based on data about recent inputs, such as observed workloads. Recent years have seen ML-based enhancements, or outright replacements, for core components such as data structures Kraska et al. (2018); Ding et al. (2020a); Galakatos et al. (2019), query optimizers Marcus et al. (2022; 2019); Van Aken et al. (2017), memory allocators Yang et al. (2020), and schedulers Mao et al. (2019); Holze et al. (2010). At the same time, the use of machine learning introduces well-known security vulnerabilities, such as adversarial inputs that cause misprediction Szegedy et al. (2013); Goodfellow et al. (2014); data-poisoning

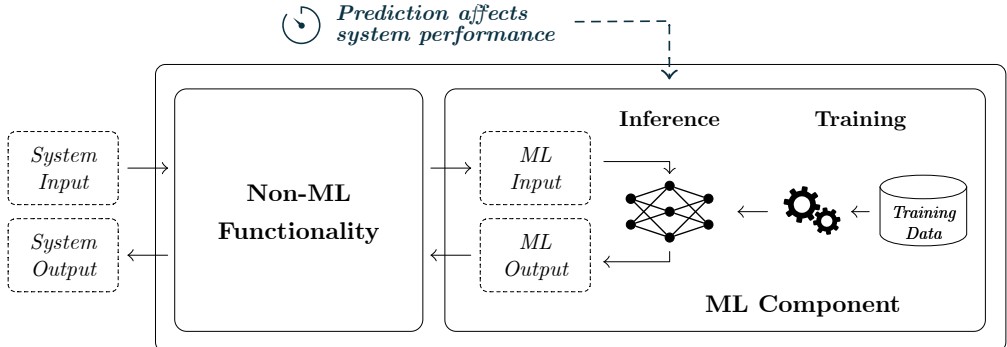

Figure 1: Learned database systems use ML models whose predictions are used only internally, yet indirectly affect externally-observable performance metrics.

attacks Chen et al. (2017b); Yang et al. (2017); Shafahi et al. (2018); or training-data leakage Shokri et al. (2017); Song & Shmatikov (2019).

In this paper, we study how the incorporation of ML exposes database systems to attacks on privacy and availability. This is rooted in the facts that, first, fitting a model to historical data distributions *increases information flows* within the system; second, ML models are often trained for average-case objectives, and have *reduced robustness* to pathological input cases. To assess the security threats introduced by the incorporation of ML, we give a simple framework for identifying categories of vulnerabilities which could logically arise when including learned components into a system: adversarial inputs and data poisoning can target the system's availability/performance. These are carefully crafted data points designed to mislead a model at test time respectively malicious points injected into the training data to affect model performance. Furthermore, timing attacks can be used to leak sensitive information about a models training-data. To examine whether these vulnerabilities lead to attacks, we survey a broad set of prominent classes of learned components being actively researched. We examine how adding ML increases a systems' attack surface, and how this could conceivably be exploited.

To validate our framework, we perform three case studies of building proof-of-concept exploits for the attacks we identified. Each exploiting a widely-cited learned component published in a top venue: the BAO query optimizer Marcus et al. (2022), and the key-value index structures ALEX Ding et al. (2020a) and PGM Ferragina & Vinciguerra (2020). We demonstrate that the conceptual vulnerabilities identified in our survey can indeed manifest in real systems.

For BAO, which is used in production by Microsoft Negi et al. (2021a), we show the attacker can reliably infer that a specific query was executed by observing its current runtime. In other words, the very success of BAO — using ML to minimize execution time of previously observed queries — introduces an information leak. This demonstrates how leakage naturally occurs in learned components that act as a shared resource by multiple clients, and therefore adapt themselves to the observed workload by all clients. This is similar to how a shared cache adapts itself to the memory workload of all operating-system processes (and users), regardless of process-memory isolation.

We analyze the ALEX index structure, which presents an improved average-case performance over traditional non-learned baselines but has no worst-case guarantees. We discover a class of pathological inputs cases which cause exponential memory blowup, and use them to crash the system within seconds when running on a 16GB RAM machine, or in about 10 minutes when running on a server equipped with 512GB RAM. This illustrates the dangers of optimizing for average-case performance and disregarding edge cases that can be maliciously invoked.

Next, we consider the PGM index, which has excellent performance *and* amortized worst-case guarantees on all operations. Worst-case guarantees can mitigate attacks on availability, yet we show that they still do not protect against privacy attacks, by demonstrating that an attacker can infer information about the distribution of victim keys by querying *other* keys.

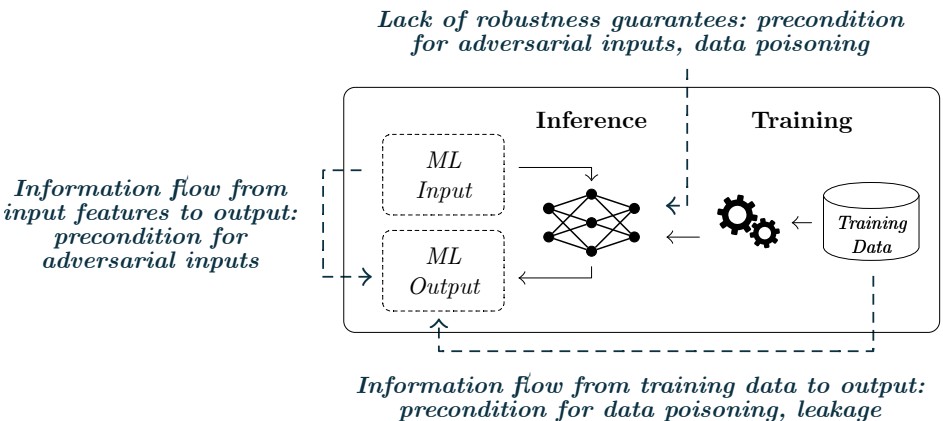

Figure 2: Information flows and lack of robustness are preconditions for ML vulnerabilities.

Lastly, we discuss mitigations. Some attacks might be prevented by careful use of existing defenses against adversarial ML, such as differentially-private training Abadi et al. (2016); Song et al. (2013) or robust training Xie et al. (2019); Chen et al. (2017a); Cohen et al. (2019). However some of the information-leakage we observe are inherent to the very quality that makes ML desirable, namely, the ability to adapt to observed input patterns.

***Contributions.*** We make the following key contributions.

- We systematize the study of the security of learned components in ML-enhanced database systems. We build a framework (Section 2) to identify and analyze ML vulnerabilities in these systems, which are corollaries of increased information flows and lack of robustness. We survey (Section 3) prominent classes of learned components and explain how these vulnerabilities are exploited in context.

- We validate our framework by demonstrating 3 exploits for attacks it identified: first, BAO (Section 4), a query optimizer that learns from past query executions and thereby allows attackers to identify which queries were executed; Second, ALEX (Section 5), a learned index structure that has excellent average-case performance but can be crashed in seconds using a meticulously-crafted series of inputs; finally, we consider PGM (Section 6), an index structure that improves on ALEX by providing excellent worst-case guarantees. Despite this best effort, PGM still exhibits a vulnerability compared to traditional approaches: PGM leaks, through query-latency measurements of the attacker's own stored data, information about underlying distribution of other stored keys.

- We discuss potential mitigations, explain why some attacks are inherent to learned components that are shared between multiple mutually-distrusting users, and identify significant gaps in current research as well as in the security practices of designing learned systems.

As learned components become increasingly adopted in practice, where they handle real-world sensitive data and their performance is critically relied upon, there is an urgent need to understand their risks. Our work provides an important step towards this goal.

## 2 A Framework

In this work, we define *learned systems* as systems that uses ML components internally to improve performance metrics such as runtime or memory usage without affecting the system's output (cf. Figure 1). In contrast, other ML-based software employs ML to directly influence outputs. For instance, approximate query processing Li & Li (2018); Chaudhuri et al. (2017) use model predictions to compute and return approximate results directly to the user. Conversely, in a learned index structure (§5, 6), the learned component predicts the location of the key in the index, only to help locate its record, which is the system's output. Model accuracy does not affect the output, only the answer's latency.

***ML increases systems' attack surfaces.*** Learned systems inherit key weaknesses of their underlying ML components — (1) lack of robustness to adversarial manipulation, and (2) increased information flows between ML inputs and outputs. The latter occurs due to the addition of (possibly multiple) training steps, as well as the fact that deep ML model architectures are designed to be able to take extraneous features as input and perform (learned) feature selection LeCun et al. (2015). Figure 2 illustrates these issues.

Two key features of learned systems exacerbate these weaknesses. First, *the line between ML and the system is often blurry.* The abstraction in Figure 1, where a single well-delineated model provides a prediction to an encapsulating system, is an oversimplification. Learned systems can include many sub-models that work together and are inter-dependent, to the point where the system's entire internal object hierarchy, architecture, and memory footprint become subject to the learned (and potentially malleable) data distribution. These levels could even interact across abstraction boundaries—for example, a learned database index accessed by a process using a learned memory allocator.

Second, *learned components are often shared resources.* Production databases have flexible access control permission systems MongoDB (2024b); MSSQL (2024) to support multiple users and roles with varying access-privilege levels; however components such as the underlying index structure MongoDB (2024a), query optimizer, or configuration knobs Xu et al. (2015) are shared between all users. When these components are learned from multiple users/roles' data and serve them at the same time, privilege-level information compartmentalization breaks.

***Some age-old systems are learned.*** An astute reader might have noticed that the defining characteristic of learned systems, i.e. adapting to observed input patterns in order to improve performance, is not new to the deep-learning era. In fact, it may precede neural networks altogether, as several "classical" hardware and software components such as branch predictors or caches contain components that adapt to input patterns. We argue that it is no coincidence that these same microarchitectural elements have been targets to countless timing attacks Yarom & Falkner (2014); Tromer et al. (2010); Osvik et al. (2006) — they carry the very risks we discuss in this paper. Importantly, our paper's focus is the growing trend of incorporating modern ML in database systems. We see a future where instead of a few well-known shared resources that adapt to input patterns across multiple users, more and more systems incorporates this principle, thereby becoming more vulnerable. Our goal is therefore to systemize our understanding of this trend.

## 2.1 Identifying attacks

Our observation about the root causes of security risks in learned systems will be the starting point for a methodology for identifying attacks on learned components in database systems. The methodology is illustrated in Figure 3, and described next.

***Step 1: Identify ML components, their added information flows, and robustness.*** It is crucial to understand the ML components of the learned system, focusing especially on the information flows that are added, and the resulting system's *worst-case* performance bounds. Here, *information flows* refer to the paths through which input data or workload observations influence system behavior via ML models—introducing new dependencies that may not exist in traditional systems.

***Step 2: Consider ML attacks on components.*** Armed with an understanding of how the ML components interact with the rest of the system, the next step is to identify which, if any, preconditions exist for attacks on the ML components of the system. In this paper, we will focus on three kinds of attacks. They all exploit new information flows introduced by ML in learned systems, or the lack of worst-case guarantees provided by the ML component. Figure 2 illustrates how information flow and lack of robustness lead to these vulnerabilities.

*Adversarial inputs.* These are inference-time inputs crafted to make a model produce attacker-chosen or erroneous outputs Goodfellow et al. (2014); Szegedy et al. (2013). A precondition of these attacks is an information flow from an attacker-supplied input to output, or that the ML component is not robust to adversarially-crafted inputs.

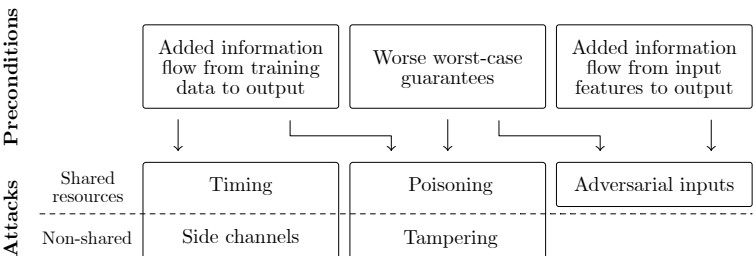

Figure 3: Our methodology for surfacing learned systems' vulnerabilities to new attacks identifies (1) whether a system is used as a shared resource between multiple participants, (2) which information flows are added, and (3) if the system has worsened worst-case guarantees as a result of using ML. Then we consider the appropriate attacker models.

*Data leakage.* Leakage of the ML component's training data can be inferred from the model's outputs Shokri et al. (2017); Song & Shmatikov (2019). In particular, membership inference attacks Shokri et al. (2017) can infer whether a given input was used to train a model. A precondition of this attack is that there is an information flow from the training data to the component's output.

*Data poisoning.* Finally, the ML component could be vulnerable to data poisoning attacks Chen et al. (2017b); Yang et al. (2017); Shafahi et al. (2018); Schuster et al. (2020a); Wallace et al. (2020b) if the training data can be influenced by the adversary to cause mispredictions. Data poisoning could arise if the component's inputs can be chosen by the attacker, and the ML's predictions can be arbitrarily skewed (i.e., lack of robustness).

**Step 3: Lift component attacks to system attacks.** The final step, after identifying the preconditions for vulnerabilities of the ML components, is to understand how attackers can exploit these vulnerabilities in context. Attack goals could be either to undermine the availability of the learned system, or violate the confidentiality of a victim by inferring protected information on their data. In this work, there are two attack models we will focus on.

*Concurrent user attackers.* For shared-resource learned systems, we consider a multi-user setting where the attacker is employing the system concurrently with the victim. We assume, here, that the system exposes an interface to both the attacker and the victim, and that standard access control is in place to prevent the attacker from accessing victim data directly. We consider three types of attackers on shared-resource learned systems: *poisoning attackers* who issue requests undermine the availability of the system by altering its state, *adversarial-input attackers* who undermine system availability by dispatching requests that cause large prediction errors in its internal model inference procedure, and *timing attackers* who infer information through query latency.

*External attackers.* Third parties who are not using the system can still leverage the attack surface that ML exposes, though they must do this indirectly, through either *tampering attacks* on the training procedure or data (e.g., externally affect the reward-signal measurements, see Section 3.4) or by utilizing a *side channel attack* to infer training-data properties. Tampering attacks can have the same goals as poisoning or adversarial-input attacks, and side-channel attacks can have the same goals as timing attacks, except that tampering/side-channel attacks are without direct access to dispatch requests to the system. Tampering and side channels correspond to strictly stronger threat models than poisoning/adversarial-inputs or timing attacks; when both are possible.

## 2.2 Traditional adversarial ML is not sufficient

Attack models of adversarial machine learning typically target ML-as-a-service systems, where the attacker directly interacts with the model, or crowd-sourced training data, malleable even by weak attackers. This does not model learned-systems attackers, due to the following distinctive characteristics.

***Label-less black-box attacks.***    Attackers often only hold black-box access to learned systems. On the face of it, this is similar to the case of (well-studied) "label-only" or "decision-only" attackers on ML-as-a-service. In contrast, however, learned-systems attackers do not even directly see model decisions. This makes traditional black-box approaches such as gradient estimation or building a surrogate model  Zhao et al. (2019); Chen et al. (2020) not directly applicable.

***Limited malleability of model inputs.***    Attackers' effect on ML-model inputs is indirect. Adversaries can only control system inputs (which are distinct from model inputs, see Figure 1) or externally tamper with it to modify model inputs.

***Limited malleability of training data.***    When an attacker can affect a learned system's training data, they usually cannot directly modify it or add data points to it. To illustrate this, consider a reinforcement-learning learned system that improves by training on data from its past executions. Reinforcement learning is particularly natural in learned systems, because the performance under the metric that the model is meant to optimize through its predictions, can be measured by the system post-hoc, i.e., after the model has already made its prediction Marcus & Papaemmanouil (2018); Krishnan et al. (2018); Mao et al. (2016). On one hand, on-line reinforcement learning exposes the system to poisoning attacks as long as untrusted entities can affect the system's inputs (or tamper with it to affect model inputs). On the other hand, the fact that the reward signal is measured by the system itself can pose a challenge for attackers who can only control its inputs and try to poison the model.

## 3    Learned-Database Systems Security Survey

Next, we want to apply the framework to several prominent classes of learned components in database systems. Figure 3 illustrates our methodology, and Table 1 summarizes the findings. Crucially, we analyze the risk of adversarial ML not in vacuum, but within the context of the system where the ML models are deployed, and we exclude attacks for which the use of ML does not increase the attack surface.

### 3.1    Learned query optimizers

Databases must determine and execute an *execution plan*, which is a sequence of basic operations—such as join, sort, or filter—that computes the result of a given *query*. A *query optimizer* is a component responsible for generating an execution plan that minimizes resource usage, for example on memory, I/O, or CPU. Plans are typically expressed as computation trees where nodes are operations and leaves are relations in the database. Query optimization is challenging because the optimizer must choose the best-performing plan without actually executing any plan. Specifically, optimizers must estimate the performance of computation tree nodes before their inputs, or their cardinality, are known. Consequently, popular optimizers are the product of massive engineering effort, with many carefully-crafted heuristics for estimating node-input cardinalities, choosing between different operation implementations and selecting operation order.

Query optimization can also be approached as a *learning* problem: as execution plans are carried out, the system gathers performance data. This data can then be used to refine and enhance future execution plans. By continuously learning from experience, the optimizer can dynamically adapt to changing workloads. Online-learned models are becoming increasingly popular Wang et al. (2024); Krishnan et al. (2018); Marcus & Papaemmanouil (2018); Marcus et al. (2019; 2022); Park et al. (2020).

***Vulnerability preconditions met.***    Learned query optimizers learn from past-query experience to optimize the latency of future ones, thereby adding information flow from the former to the latter.

***Attacks that become possible.***    When databases are used by multiple users with varying privilege-levels, their shared query optimizers are at risk of timing attacks that reveal information on past queries, and data poisoning that degrades the performance of future queries. This implies that *tampering* and *side channels* are also possible (see Section 2.2). In Section 4, we demonstrate proof-of-concept of a timing attack against BAO Marcus et al. (2022), a state-of-the-art learned query optimizer.

Table 1: Overview of attacks that become likely due to use of ML in internal components. We could exclude very few attacks, implying that these issues are universal in learned systems.

| Learned system | Adversarial inputs | Training-data leakage | | Adversarial training data | |
|---|---|---|---|---|---|
| | | Timing | Side channels | Poisoning | Tampering |
| Query optimization [a] | | ✓ | ✓ | ✓ | ✓ |
| Index structures [b] | ✓ | ✓ | ✓ | ✓ | ✓ |
| Learned configuration [c] | | ✓ | ✓ | ✓ | ✓ |
| Workload forecasting [d] | ✓ | ✓ | ✓ | ✓ | ✓ |

[a] Wang et al. (2024); Park et al. (2020); Krishnan et al. (2018); Marcus & Papaemmanouil (2018); Marcus et al. (2019; 2022)
[b] Kraska et al. (2018); Nathan et al. (2020); Ding et al. (2020b); Kipf et al. (2019); Abu-Libdeh et al. (2020)
   Wei et al. (2020); Ding et al. (2020a); Ferragina & Vinciguerra (2020); Galakatos et al. (2019)
[c] Van Aken et al. (2017); Kanellis et al. (2020); Ma et al. (2018)
[d] Ma et al. (2018); Elnaffar & Martin (2004); Holze et al. (2010; 2009)

## 3.2 Learned index structures

Index structures are key-value stores designed to support key queries, range queries, insertions, and deletions. Common implementations of these structures are based on B-trees, which are height-balanced trees where each node has a limited number of children, and the keys are stored in sorted order in the leaves.

Kraska et al. (2018) proposed a novel perspective on index structures by viewing them as cumulative distribution functions (CDFs) that map keys to their location. They suggested using optimization techniques for continuous functions, such as gradient-based or convex optimization, to fit a CDF model to perform lookups. The resulting approaches Kraska et al. (2018); Nathan et al. (2020); Ding et al. (2020b) have shown superior performance compared to traditional index structures, particularly on read-only benchmarks. Specifically, compared to B-trees, these learned index structures offer faster reads and require less space, which is especially beneficial for access over networks Abu-Libdeh et al. (2020); Wei et al. (2020). More recent approaches also support updates Ding et al. (2020a); Ferragina & Vinciguerra (2020); Galakatos et al. (2019).

***Vulnerability preconditions met.*** Learned index structures have an information flow from the keys stored in the dataset (the training data) to the location of the record (the prediction) and the accuracy of the prediction directly impacts the lookup time. Kraska et al.'s recursive model index (RMI) Kraska et al. (2018) and similar subsequent approaches do not offer robustness guarantees, making them less reliable compared to traditional index structures. Among newer methods that support updates, PGM Ferragina & Vinciguerra (2020) includes robustness guarantees for all operations, but ALEX Ding et al. (2020a) and FITing-Tree Galakatos et al. (2019) have exponentially-worse bounds in certain insertion scenarios and, thus, reduced robustness compared to conventional approaches.

***Attacks that become possible.*** Index structures underlie many common databases MongoDB (2024a) that can be shared by multiple users with varying privileges. Due to the implicit information flow from training data to the model's predictions, learned index structures are exposed to both data poisoning attacks that degrade their performance and timing attacks that snoop on data of victim users. Kornaropoulos et al. (2020) demonstrated a data poisoning attack that degrade the (read-only) RMI index' performance; in Sections 5, we show that ALEX, a newer write-enabled index, is even more vulnerable. Our attack causes an out-of-memory crash within seconds using a sequence of adversarial insertions. In Section 6, we further demonstrate a timing attack on the PGM index, revealing that even data structures with excellent worst-case performance can be exposed to attacks that their non-learned counterparts are immune to.

## 3.3 Learned configuration

Databases have hundreds of "knobs", i.e., parameters which control aspects such as the amount of caching memory or how frequently data is flushed to storage. Tuning these parameters is a complex and time-consuming task, yet it lends itself well to automation by replaying past workloads, measuring performance,

and using the measurements to guide parameter-space search. Consequently, ML-based approaches are increasingly popular for parameter tuning Van Aken et al. (2017); Kanellis et al. (2020); Ma et al. (2018).

***Vulnerability preconditions met.*** Optimal parameterization can depend on (1) the stored data, (2) the workload, and (3) the hardware and software stack. As a result, during automated tuning, a potential information flow is added from each of these onto virtually any decision that depend on parameterization.

***Attacks that become possible.*** Stored data and workload may leak between different users in a system via timing measurements (i.e., timing), or be poisoned by adversarially-crafted workloads to affect these metrics (i.e., data poisoning). Perhaps more dangerously, however, learned configurations are highly susceptible to tampering attacks that adversarially affect the hardware/software stack characteristics learned during tuning. For example, a process on the tuning machine may intentionally consume more memory than normal during tuning, so that tuning sub-optimally optimize for low memory usage. Similarly, physical adversaries can try to compromise the hardware during tuning (e.g. cause faults Boneh et al. (1997)). Notably, as opposed to traditional tampering attacks that invoke computation or memory faults that are unintended implementation artifacts and physical effects Boneh et al. (1997); Gruss et al. (2016), here, the tuning procedure is actively *trying* to learn the environment's characteristics, which can be manipulated by an attacker.

### 3.4 Workload forecasting

Workload forecasting predicts future workloads based on recent patterns, enabling the database to adjust its configurable parameters accordingly Ma et al. (2018); Elnaffar & Martin (2004); Holze et al. (2010; 2009). For example, Ma et al. (2018) proposed to group queries to a database with highly correlated appearances into clusters and then predict the short-term and long-term workloads for these clusters.

***Vulnerability preconditions met.*** Workload forecasting introduces an information flow from past requests (training data) to the database performance metrics (output).

***Attacks that become possible.*** Due to the added information flow, workload forecasting may introduce vulnerability to data poisoning and data leakage in multi-user settings. For example, in a database system, workload forecasting may result in a construction of a temporary column index whenever the column is expected to be heavily used. An attacker who can submit queries that use this column and observe reduced latency whenever its index exists might be able to infer when the column is heavily used by others.

### 3.5 Discussion

These areas discussed so far represent the most prominent directions we identified in the research on learned database systems, with each being supported by multiple papers. Table 1 summarizes our findings. In almost all systems, the attack surface increases for almost all attack types of interest. For shared-resource systems, we were able to exclude increased risk for only a few adversarial-input attacks, for two reasons: first, because attackers who issue requests to query optimizers can overload these systems without attacking the ML component, by simply submitting actually-resouce-intensive queries. Second, because learned configuration produces knob values that are unlikely to be any less robust than those produced by manual tuning.

> **Takeaway:** The incorporation of ML universally compromises shared-resource systems to adversarial inputs, timing, and poisoning.

In the upcoming sections, we validate our findings by showing that exploiting the increased attack surface is indeed feasible. To this end, we implement and evaluate proof-of-concept attacks in Sections 4 through 6. The targets for these attacks were selected conservatively: we focus on work that was published in top-tier venues and had open-source implementations, and select attack goals that are non-trivial and where empirical evaluation carries actual value (for example, poisoning the BAO query optimizer by issuing the same query enough times will, by BAO's design, completely eliminate the value of the learned component; we therefore focus on evaluating a timing attack instead).

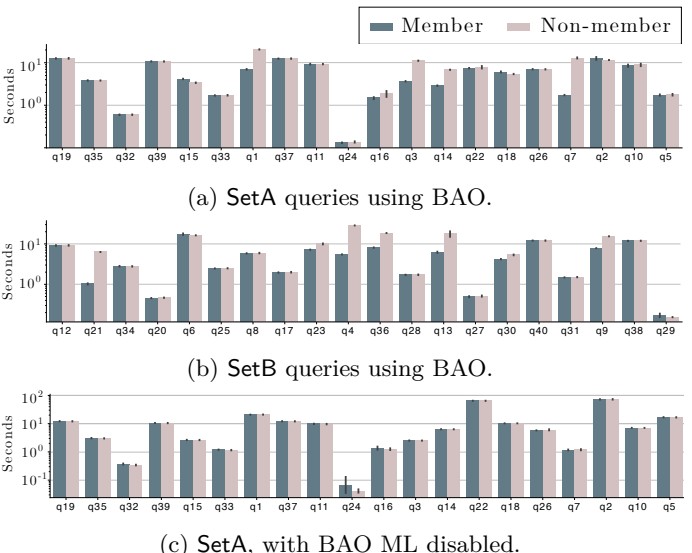

(a) SetA queries using BAO.

(b) SetB queries using BAO.

(c) SetA, with BAO ML disabled.

Figure 4: Query-answer latency for member and nonmember queries with BAO enabled and disabled for two disjoint 20-query sets, SetA and SetB. For many queries, execution time differs significantly depending on whether the query was part of the model's training set, revealing a timing side-channel. With BAO disabled, this distinction disappears, confirming the leakage is due to the learned model.

## 4   Case Study: The BAO Query Optimizer

We start by demonstrating query leakage *i.e.* whether a particular query was executed by the database in the past $N$ queries. To this end, we target the online-learned query optimizer BAO Marcus et al. (2022).

***Background: BAO.*** Recall that for traditional query optimization, different execution plans are considered to minimize resource load. BAO introduces ML to perform this selection Krishnan et al. (2018); Marcus & Papaemmanouil (2018); Marcus et al. (2019). In Postgres, each execution plan, known as an "arm", is produced deterministically with one set of configuration parameters that include whether to enable merge join, loop join, etc. BAO improves the conventional query optimizer by selecting "arms" that steer it to produce better plans. Therefore, BAO uses a variant of Thompson sampling Thompson (1933) to balance exploration and exploitation of its experience. Simply put, BAO retrains every $N$ queries, using the $N$ queries as the training set.

***Observing and characterizing the leakage.*** We first assess potential leakage by performing the following experiment. We use the Cardinality Estimation Benchmark (CEB) Negi et al. (2021b), a 13K-query benchmark generated from 16 templates by mutating filter predicates. As far as we are aware, this is similar to the dataset on which BAO was originally evaluated by Marcus et al. (2022) (the authors did not respond to our email inquiry). We use the 40-query subset of CEB used for training and benchmarking in Marcus et al. (2022). We divide the queries into two disjoint 20-query sets, SetA and SetB. For each set, we measure each query's latency in two scenarios, member and nonmember, such that in member the query is in the training set, and in nonmember it is not. The former simulates the case where the model has already seen the queries, whereas the latter simulates a scenario where it has not. We expect the query latency to be lower in the member scenario, suggesting that past experience information (i.e., the training set) leaks via query latency. To isolate the effect of BAO's learning from potential effects such as Postgres' "native" optimizations like query or computation caching, we repeat this experiment but with BAO disabled.

Figures 4a and 4b depict the results of BAO-enabled experiment for sets SetA and SetB respectively. For many queries in the BAO-enabled databases, latency in the nonmember scenario is often much larger than that in the member scenario (note the log scale), suggesting that the attacker can distinguish between the set using a single query. Notably, the difference is larger for slower queries that naturally require more time

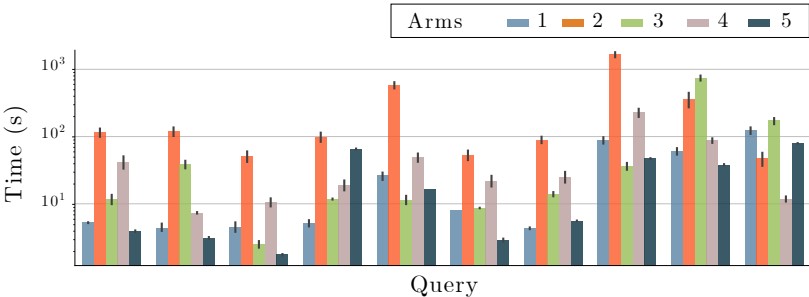

Figure 5: Query-answer latency of different arms for 10 sampled complex queries. For each query, the latency is very distinct across different arms.

to execute, implying that slower queries are more vulnerable to past-query inference attack. Figure 4c shows the results for SetA when BAO is disabled, there is hardly any difference in the execution time between member and nonmember, suggesting that, indeed, the leakage is due to ML. The results for SetB are similar and not reported for conciseness.

***Complex queries leak via latency measurement.*** In Figure 4a and 4b, leakage is more noticeable for queries with longer execution time. These queries are often complex and take more time to execute even with an optimized execution plan selected by BAO. Consequently, the latency gap between member and nonmember setting is well above several seconds. This would allow the attacker to first estimate the latency of a query by training a shadow model replicating BAO, and then compare this latency estimate with the actual latency of execution on the deployment of BAO being attacked. Furthermore, we observe that for such complex queries, different arms lead to distinct latency values, allowing us to infer which arm was chosen by BAO based on the latency. We show latency measurement of 10 queries sampled from the set of complex queries in Figure 5 with each of the query executed 10 times and the 95% CI for each arm shown. Clearly, the runtime for each query among the five arms are very distinct with both large absolute gaps (recall the log scale) and non-overlapping CIs. This observation enables us to confidently determine the arm chosen by BAO based on the query execution latency.

***Attacker model.*** We envision an attacker whose goal is to determine if a *target query* was recently executed (and thus included in the last $N$-query training batch). This attacker is capable of making queries to the database using BAO, and observing their execution time. Furthermore, we assume the attacker knows the details of the learned system (including the data and different types of arms) but not the received queries.

***Attack method.*** Our attack performs the following steps: ❶ In an off-line phase, record the execution latency for each query with different arms by executing the query multiple times, forming a mapping between arm and execution latency for each query. ❷ In an off-line phase, for each query, characterize the distribution of arm selection by BAO by training it with various datasets. ❸ Choose a target query. ❹ Dispatch the query to the BAO-enhanced database. ❺ Measure the query execution latency, and determine which arm was chosen by looking up the mapping developed in ❶. We assume our attack can measure execution latency with 1-second accuracy (which is highly plausible even for a remote adversary). ❻ Depending on the arm BAO chose, we can either be certain it was recently evaluated, be certain it was not recently evaluated, or quantify the precision and recall of making such predictions. We note that ❶ requires that the execution latency between arms for a particular query are distinct enough. Our empirical results from Figure 5 confirm that this is a reasonable assumption.

***Experimental setup.*** We now perform a simulation of our attack scenario. We assume BAO trains using $N = 100$ queries. We select 167 target queries from CEB by choosing queries whose execution plan contains 16 join operations, which are considered fairly complex Leis et al. (2015). We construct a *benign training set* by randomly sampling 100 queries (benign queries) from CEB, and an *adversarial training set* by replacing 1 randomly-chosen benign query with one of our target queries. We train BAO on the benign set and the adversarial set, and repeat this entire process for each target query 50 times with randomness over both

Table 2: Recall and query distribution across different precision levels for both member and nonmember settings. Precision groups reflect the confidence level with which the attack determines whether a query was part of BAO's recent training set. For each group, we report two key metrics: the proportion of total queries that fall into the group and the average recall—that is, the frequency with which the classification was correct. Our attack achieves relatively high precision while maintaining satisfying recall.

| Setting | Precision Group | 0.5-0.6 | 0.6-0.7 | 0.7-0.8 | 0.8-0.9 | 0.9-1.0 |
|---------|-----------------|---------|---------|---------|---------|---------|
| member | Average Recall | 0.56 | 0.54 | 0.37 | 0.28 | 0.24 |
| | % of Queries | 22% | 24% | 17% | 12% | 25% |
| nonmember | Average Recall | 0.29 | 0.33 | 0.37 | 0.38 | 0.15 |
| | % of Queries | 2% | 11% | 16% | 24% | 47% |

benign queries and BAO initialization. After training BAO 50 times for each target query, we obtain an arm distribution for both member and nonmember setting. We then execute the target query on the victim BAO model and identify the arm it chose based on the mapping between arm and latency obtained in the off-line setting. We aim to maximize the precision of our predictions for both member and nonmember cases.

***Results.*** We calculate the precision and recall of each target query for both of the member and nonmember settings. Specifically, for each query, we find the arm that leads to maximum precision classifying the two settings. By doing so, we emphasize precision for each setting since being able to confidently determine either member or nonmember setting with some sacrifices of recall is acceptable by the attacker. Since we focus on precision, we divide target queries into five precision groups and plot the portion of target queries falling into those groups as well as the average recall for them. In Table 2, it can be seen that many target queries belong to high precision group especially for nonmember setting with more than 40% having precision between 0.9–1.0. The corresponding average recall for these precision groups are also satisfying. For member setting, average recall decreases as precision group gets higher whereas in nonmember setting, average recall is the highest for precision group 0.8–0.9. Altogether, these results demonstrate how sensitive information (the past execution of a query) may leak due to the introduction of ML in the query optimizer.

> **Takeaway:** Sensitive information can leak in shared-resource systems due to the introduction of ML.

## 5 Case Study: The ALEX Learned Index

Next, we examine a distinct characteristic of learned systems: optimizing for average-case performance can unintentionally lead to catastrophic outcomes in worst-case scenarios. To illustrate this, we consider an attack against the ALEX learned index.

***Background: ALEX.*** ALEX is a learned index structure that supports lookups, range queries, insertions, bulk insertions, and deletions Ding et al. (2020a). The ALEX structure is organized in a tree hierarchy where each node can store a range of keys. Sibling nodes' ranges are disjoint; internal nodes point to an array of nodes sorted by their ranges; leaf nodes point to a sorted array of keys. Each node contains a linear function (of the form $f(k) = ak + b$, with $a, b \in \mathbb{R}$), that maps a key to its containing-range sub-tree (for internal nodes) or its position (for leaf nodes). Lookups are performed by following the linear models with predictions rounded to the nearest integer since array positions are discrete. In case of prediction errors, exponential search is used to find the correct position in the sorted array, starting from the predicted position.

*Insertion internals.* ALEX optimizes for fast insertions and therefore includes redundant positions for key values and internal-node pointers. The first type of redundancy is the use of gapped arrays in leaf nodes. These allow quick insertions into locations where there is a gap; when a key needs to be inserted within a contiguous sequence of keys, ALEX shifts keys toward the nearest gaps to create space. This operation can be costly in the worst case, where the number of shifts equals the number of keys in the node. However, there are even worse scenarios discussed next.

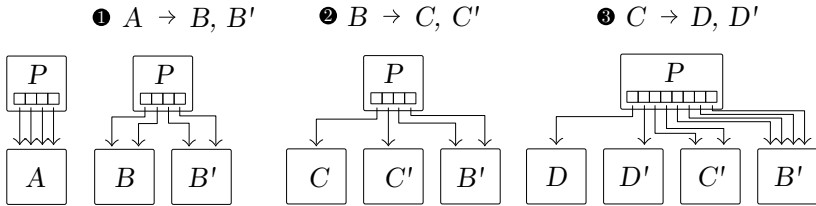

Figure 6: Repeated sideway splits to a subtree's leftmost node cause exponential memory blowup. First, node A, child node of $P$ with 4 redundant pointers, is split into $B, B'$ (Step ❶), then B to $C, C'$ (Step ❷), then $C$ to $D, D'$ (Step ❸). On the last split, there are no redundant pointers into $D$, and more memory is allocated by doubling $P$'s pointer array size.

Specifically, when a node reaches its upper density bound (i.e., becomes too full for further insertions), or when ALEX's internal heuristics determine that the node's lookup or insertion performance is suboptimal due to overpopulation, ALEX performs a split. The preferred method is a *sideways split*, which requires a second type of redundancy: duplicate pointers. In this context, internal nodes' pointer arrays contain duplicate pointers to the same child nodes. During a sideways split, a child node with duplicate pointers is divided into two new nodes, each covering half of the original node's key range. The duplicate pointers in the parent node are then split between these new nodes. This method ensures that internal nodes do not need to be retrained because the linear model mapping keys to subtrees remains accurate after the split. If no duplicate pointers exist during a sideways split, ALEX allocates more pointers by doubling the size of the parent node, using the extra slots to double the number of doubled pointers to each of each child nodes.

Once the parent array reaches a maximum size, the parent can also perform a sideway split. Only when *all* predecessors' sizes reach the maximum size, ALEX will perform a *downward split*. In a downward split, a split node's keys are reallocated into two new nodes, and the original node converts into an internal node whose children are the two new nodes. Downwards splits are straightforward and do not require any redundancy but they increase the tree's depth. For this reason, downward splits are only used as a last resort when a sideways split is no longer possible.

***Worst-case trade-off.*** The splitting mechanism can lead to *exponential memory growth*. Consider the scenario illustrated in Figure 6. Here a child node $A$ has four duplicate keys into it, and sideway-splits due to overpopulation into new nodes $B$ (on the right) and $B'$ (on the left), and then $B$ sideway-splits again into $C$ and $C'$, again on the right and left, then $C$ can no longer split sideway unless the parent node array doubles its size to allocate a duplicate pointer into $C$. We observe, that if $C$ now splits again into $D, D'$, and $D$ into $E, E'$ and so forth, then the parent array size grows exponentially with each split, until it hits the maximal node size (for ALEX's default parameterization, 16MB).

As a result, ALEX permits a worst-case scenario where memory usage can blowup exponentially to attain good average-case performance. This occurs because the parent node is never retrained and cannot reallocate the leftmost key range to subtrees other than the leftmost child's. According to Ding et al., this choice was made because retraining is costly.

***Poisoning attack on ALEX.*** Our attack has one simple capability: *insert keys into the index*. The attacker's goal is to leverage the above-described effect to jeopardize the availability of ALEX by increasing its CPU and memory consumption. This can result in (1) much higher maintenance costs, for example because the cloud instance running ALEX must be of a higher tier, and (2) if the index is serving other users in addition to the attacker, their requests will be delayed, and (3) it can ultimately lead to a full denial of service, if ALEX crashes due to being out of memory.

We assume the attack has minimal knowledge of the key distribution the ALEX structure contains and only roughly knows the range of keys used. For the attack, the adversary then guesses a random position in the key range and start inserting keys to the left of this position. Our observation is that this causes repeated insertions to the leftmost node within a sub-tree, and therefore, as described before, exponential memory blowup due to a series of sideway splits where, for each split, the parent node doubles the size of its pointer array. For more details refer to Appendix A.1 where we prove this informally.

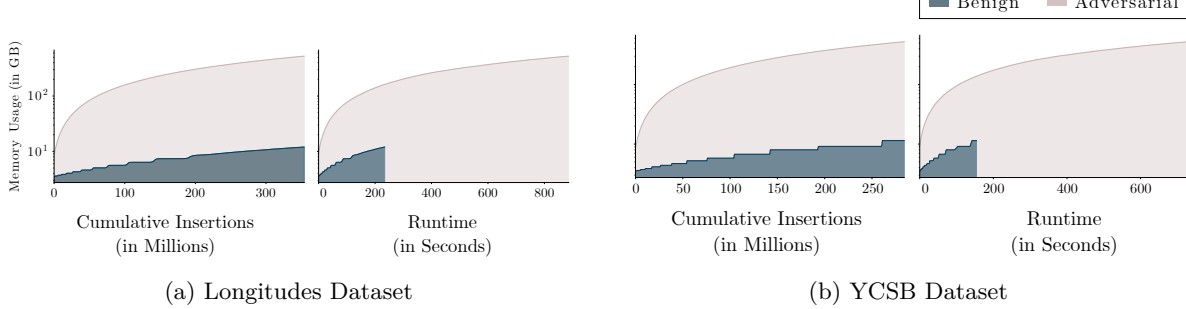

Figure 7: Memory usage as a function of number of insertions and runtime.

***Evaluation.*** We evaluate the attack on two datasets from the original publication Ding et al. (2020a). First, the *YCSB* dataset, containing user IDs from the YCSB benchmark Cooper et al. (2010), and second the *longitudes* dataset with longitudes of locations around the world from Open Street Maps. Each dataset contains 200 million records. In both cases, we first build an ALEX index for the first 10 million elements by instantiating the index and bulk-loading the keys. Then, we simulate an adversary who inserts keys using the method described above. For comparison, we also simulate a scenario where a benign user inserts non-maliciously-crafted keys drawn sequentially, similarly to the experiments from Ding et al.

We run ALEX on a server with two Intel Xeon E5-2686 CPUs and 512GB of memory. We use ALEX's benchmark program and supply a binary file containing (1) the keys to bulk-load and (2) the attacker and victim's keys for insert/lookup operations. We set up the benchmark program to read 1 million operations at a time from the file, and report their performance statistics. To prototype how ALEX would behave on a machine that is not geared with an unusually high amount of RAM, we additionally run the attack on a machine with an Intel i7-7600U CPU and 16GB of memory.

*Results.* In Figure 7a and 7b, we plot memory usage as a function of number of insertions and runtime on Longitudes and YCSB dataset. In the benign scenarios, ALEX's memory usage peaks at just over 10 GB throughout the experiment. However, under the attack, memory usage exceeds 100 GB within 136 and 107 seconds, and crashes due to out-of-memory in 15 and 12 minutes (resp.). Since the adversary's insertions require many expensive memory allocations, it slows down ALEX by a factor of almost ×4. The attack fills up ALEX's memory slightly slower for the YCSB dataset, likely due to its sparser key range. We empirically verify this in Appendix A.2.

On the machine with 16GB of memory, ALEX crashes within 14 seconds after the attack initiated, after 14 million insertion operations by the attacker.

***Detectability.*** ALEX can try to detect the scenario of a single user who dispatches many insertion operations, of which a large proportion cause node splits, and block the user. This would require breaking the standard abstractions in many systems, in which index structures are decoupled from user authentication/identity mechanisms (e.g. of a database that internally uses the index). The defense's detection threshold would then have to be carefully tuned for each deployment, depending on the expected benign workload and memory increase rate. Unfortunately, even by paying that price, this would not be sufficient: the adversary can slow down the attack as much as needed, and interleave their insertions with other operations, to stay undetected while still causing the highest memory increase rate allowed by the system. If at any point the attacker is detected, they can potentially create, or collude with, another account.

> **Takeaway:** Replacing worst-case guarantees with average-case performance is a reliable signal of a vulnerability to adversarial ML in learned systems.

# 6 Case Study: The PGM Learned Index

The attack on ALEX leverages our ability to craft insertion operations that cause a worst-case scenario, in terms of memory and time. We would not be able to cause such a catastrophic failing in a data structure that has tighter worst-case behavior bounds. The PGM index Ferragina & Vinciguerra (2020) is a dynamic learned index that supports updates and insertions, while being meticulously engineered to bound worst-case (amortized) complexity, in both memory and time. For example, index construction works in linear time, membership queries work in squared-logarithmic time, and insertions work in logarithmic amortized time. It is tempting to deduce that, due to its excellent worst-case behavior, the PGM index will be spared from ML vulnerabilities.

In this section, we explain and demonstrate that the very property that makes it so performant — namely fitting the index structure to the key distribution — inherently introduces information leakage when a PGM index is used by multiple users whose data is meant to be segregated (e.g., users of a database, see Section 3.2). PGM's worst-case guarantees are provided for availability without consideration of the risk for privacy leakage. Our results show that it is possible for mechanisms that provide worst-case guarantees to still introduce new vulnerability dimensions for security through runtime differences.

***Background: the PGM index.*** At the core of PGM is a linear-time streaming algorithm that finds the optimal piecewise-linear function which approximates, up to an additive error parameter $\epsilon$, the key-to-index mapping for a (sorted) array of keys. PGM applies this algorithm to build a multi-level index, where each level is a piecewise-linear approximation of the data on the next level, and the penultimate layer approximates the actual key-to-index mapping. This structure does not contain gaps like ALEX, and additional machinery is required to support efficient insertions (see Ferragina & Vinciguerra (2020)).

Since each level's linear model is an $\epsilon$-approximation of the key positions in the level below it, lookups are performed by traversing the index from top to bottom, getting the model's prediction $p$ for the key's position, and performing a binary search for the key's actual position within $[p - \epsilon, p + \epsilon]$. This implies that the number of operations performed during a lookup depends only on $\epsilon$, which is a constant parameter regardless of the data, and the index's height.

***Attack: distinguishing two key distributions.*** Indexes are often underlying databases serving multiple users with varying privilege levels MongoDB (2024a;b), where it is required that users' data is properly compartmentalized. We will show that, simply by measuring query latency of accessing *their own keys*, an attacker can easily tell between two different victim key distributions, due to the index size being dependent on them, dramatically affecting attacker's latency.

***Experimental setup.*** We load datasets of equal size, drawn from two key distributions, onto PGM's open-source implementation: A, a normal distribution with mean $\mu_1 = 5e6$ and standard deviation $\sigma_1 = 1e6$, and B, a mixture of 50 normal distributions with $\sigma_2 = \sigma_1/50$ and means $\mu_2 \in \{0, 2\mu_2, ..., 50 \cdot 2\mu\}$ where each of the distributions is weighted equally. We add 100 *attacker keys* to each dataset, uniformly from $[\mu_1 - 2\sigma_1, \mu_1 + 2\sigma_1)$ (within the approximate range of keys in both A and B). We make the assumption that those are the only keys the attacker is allowed to access.

We generate the datasets by sampling from the above distributions, built the index, and measure (1) index height (number of traversed levels while searching queries, akin to tree height for tree-based indexes) and (2) the attack's query latency. We repeat this 50 times for multiple values of $\epsilon$, and average the results.

We also take the same latency measurements for a popular B-tree CPP implementation B-tree (2011), and randomized insertion order when constructing the B-tree. We note that, by construction, B-trees are oblivious to key values, and only observe the key order (and insertion order). Therefore, we expect the B-tree to be far less sensitive to the differences between key distributions, as long as the datasets contain the same number of keys. We repeat this entire experiment again, this time freeing the index's memory and reallocating it between each two attacker queries, to mitigate any effects of leakage stemming from microarchitectural elements such as caches. We run our experiments on a machine with a Intel i9-9940X CPU and 128 GB of memory.

Table 3: PGM's height and attacker's latency measurements (nanoseconds) for datasets of different key distributions. Distribution B results in comparatively flatter indexes, resulting in quicker query times for PGM. Measurements are averaged across 50 experiments.

| Model | A | | B | |
|---|---|---|---|---|
| | Height | Latency | Height | Latency |
| PGM, $\epsilon =$ 8 | 3.0 | 165.14 | 3.0 | 146.52 |
| PGM, $\epsilon =$ 16 | 3.0 | 126.46 | 3.0 | 120.32 |
| PGM, $\epsilon =$ 32 | 3.0 | 134.18 | 2.52 | 113.98 |
| PGM, $\epsilon =$ 64 | 3.0 | 142.62 | 2.2 | 118.06 |
| PGM, $\epsilon =$ 128 | 3.0 | 157.64 | 2.0 | 124.3 |
| PGM, $\epsilon =$ 256 | 3.0 | 170.06 | 2.0 | 126.68 |
| PGM, $\epsilon =$ 512 | 3.0 | 183.24 | 2.0 | 135.08 |
| PGM, $\epsilon = 1024$ | 3.0 | 191.46 | 2.0 | 142.36 |
| B-tree | - | 112.28 | - | 99.32 |

Table 4: PGM's height and attacker's latency measurements (nanoseconds) for datasets of different key distributions, in the setting when we free the index's memory and rebuild it between every two queries. Distribution B results in comparatively flatter indexes, resulting in quicker query times for PGM. B-trees do not exhibit this leakage. Measurements are averaged across 50 experiments.

| Model | A | | B | |
|---|---|---|---|---|
| | Height | Latency | Height | Latency |
| PGM, $\epsilon =$ 8 | 3.0 | 204.38 | 3.0 | 181.54 |
| PGM, $\epsilon =$ 16 | 3.0 | 206.76 | 3.0 | 183.08 |
| PGM, $\epsilon =$ 32 | 3.0 | 227.32 | 2.66 | 187.84 |
| PGM, $\epsilon =$ 64 | 3.0 | 240.12 | 2.06 | 194.1 |
| PGM, $\epsilon =$ 128 | 3.0 | 270.84 | 2.0 | 219.04 |
| PGM, $\epsilon =$ 256 | 3.0 | 287.48 | 2.0 | 233.04 |
| PGM, $\epsilon =$ 512 | 3.0 | 308.64 | 2.0 | 256.92 |
| PGM, $\epsilon = 1024$ | 3.0 | 329.96 | 2.0 | 276.12 |
| B-tree | - | 185.2 | - | 185.52 |

**Results.** Table 3 summarizes the results for the first experiment. As expected, distribution A results in more levels for PGM to traverse, and correspondingly, higher query latency. There is *some* difference in latency measurements for B-trees, as well, albeit much smaller, and it stems from microarchitectural constraints and optimizations. In contrast, the leakage for PGM is *algorithmic*: PGM has to perform more operations for distributions whose functions are more difficult to approximate using its piecewise-linear approach. Table 4 shows the results for the experiment where we rebuild the index between every two queries, to mitigate operating system and hardware side-effects. PGM remains leaky, whereas B-trees exhibit consistent timing across the two distributions.

> **Takeaway:** Excellent worst-case behavior does not inoculate a system against attacks. In learned systems that are shared between multiple users, leakage is inherent to the very quality that makes ML attractive: the system's ability to adapt to observed input patterns.

# 7    Related Work

In this work, we systematically examine the security implications of integrating *learned components* into database systems and, specifically, focus on adversarial inputs, poisoning, and leakage attacks.

***Adversarial ML.***    Other types of attacks also exist, such as *distribution-chain attacks*, where the adversary directly manipulates software at its source (like manipulating weight values Gu et al. (2017); Kurita et al. (2020); Schuster et al. (2020b) or training code Bagdasaryan & Shmatikov (2020)), or *model stealing* attacks Juuti et al. (2019); Wallace et al. (2020a); Tramèr et al. (2016) that train a model to imitate an exposed prediction API. We note that these latter ML vulnerabilities are less pertinent to our investigation, as they are not products of learning per se and not logically exclusive learned models — for example, distribution-chain attacks exist for all software, and non-learned algorithms can also be "extracted" (i.e., stolen) via reverse engineering.

***Learned systems.***    Complementary to our exploration, Apruzzese et al. (2023) investigate the gap between research on attacking ML-as-a-service systems and actual attacks on deployed ML models in practice. Like our work, they consider ML models as components of a broader system. However, their focus is on the discrepancy between research scenarios and practical applications. Furthermore, Debenedetti et al. (2024) investigate privacy-related side channels created by learned components.

***Microarchitectural side-channels.***    Microarchitectural attacks such as cache attacks Osvik et al. (2006); Yarom & Falkner (2014) are a universal threat to any software that handles sensitive data, illustrating the danger of inadvertent information flow through shared resources, i.e. the same property that characterizes how ML is being used in software systems.

***Attacks on data structures.***    Crosby & Wallach (2003) observed in 2003 that many commonly used data structures are optimized for average-case performance yet have expensive edge cases that can be maliciously invoked, leading to attacks like HashDoS attack Wälde & Klink (2011). Our attack on ALEX shows how the advent of learned systems can reintroduce this risk. Other prior works considered attacks on traditional data structures, for example a data-leakage attack that recovers B-tree training data (i.e., the indexed keys) but requires writing into the index to detect node splits Futoransky et al. (2007). Kornaropoulos et al. (2020) mounted the first attack on a recently-proposed learned system, a poisoning attack on the original read-only RMI index. In a more recent and concurrent work, Yang et al. (2023) present a poisoning attack against ALEX similar to our investigation. Both works, however, did not consider other learned systems, nor systematically study the root causes of ML vulnerabilities and their manifestation across learned-systems.

# 8    Mitigations

Our core framework of thinking about learned components helps audit systems and identify potential vulnerabilities; however, it is a manual process which requires domain expertise in both ML and the system. Here, we consider other approaches.

***Worst-case complexity guarantees.***    Poor performance on edge cases can have catastrophic consequences since it can be adversarially invoked, as our attack on ALEX demonstrates. It is thus important to require that replacements of traditional systems maintain performance bounds even at the worst case. However, our attack on BAO does not rely on worst-case behavior, and our attack on PGM shows that adversarial-ML leakage is possible even with excellent worst-case bounds, so robustness is not a panacea.

***Constant-time responses.***    Timing attacks can be mitigated by adopting constant-time responses; however, this would also nullify any benefit from the use of ML that is meant to optimize response times.

***Algorithmic ML defenses.***    An extensive line of work revolves around fortifying ML against privacy leakage Abadi et al. (2016); Song et al. (2013), adversarial inputs Xie et al. (2019); Chen et al. (2017a), and poisoning Udeshi et al. (2019); Qiao et al. (2019); Liu et al. (2018); Xu et al. (2019). While certain data

leakage is inherent to shared learned systems (see Section 6), it may be somewhat mitigated by the use of, for example, differential privacy. Such solutions have expensive trade-offs, and they are never considered in work on learned systems including, to our knowledge, caches or other leaky microarchitectural elements. Therefore, the impact of this on performance remains an open question. Furthermore, separate mechanisms are required to *identify* the threats, before algorithmic approaches can be used.

> **Call to action:** The efficacy and costs of using algorithmic defenses such as differential privacy on learned systems requires more extensive research.

***Tracking information flows.*** Many ML vulnerabilities stem from increased information flows, not decreased robustness. While robustness guarantees are often clearly analyzed when constructing learned systems Kraska et al. (2018); Ding et al. (2020a); Galakatos et al. (2019); Ferragina & Vinciguerra (2020), information flows never are, so their risks remain implicit. Approaches such as taint analysis Enck et al. (2014); Arzt et al. (2014) or information-flow control Zdancewic et al. (2002); Myers (1999); Krohn et al. (2007) may be helpful in understanding ML deployment in context. Such approaches pose a burden for developers Zdancewic (2004), which may be significantly alleviated in ML by tailoring them for the programming abstractions common in ML, for example by adding security labels to (coarse-grained) processing steps or (finer-grained) Tensor variables. Combining such approaches with ML defenses that act as label declassifiers or endorsements is an entirely unstudied direction with fascinating open questions, for example "what effect should an $\epsilon$-differentially-private mechanism have on it's input's privacy label"? Exploring these questions has the potential to influence ML pipelines more broadly and are not limited to learned systems.

> **Call to action:** Work on learned systems need to analyze and make explicit added information flows and their dangers, similarly to the way worst-case bounds (or lack thereof) should be reported.

> **Call to action:** We call for information-flow analysis and control for ML abstractions like data-pipeline processing steps and Tensor variables. We envision approaches that incorporate algorithmic defenses as declassifiers and endorsements.

## 9 Conclusion

This work systematically discusses the security implications of a new and rapidly-growing trend in the use of ML in database system: employing ML models as internal system components to improve performance. We propose a framework for identifying and reasoning about vulnerabilities this practice can introduce, use it to identify potential threats to various classes of learned components, and validate it by evaluating identified attacks against three prominent ones. Our findings highlight the dangers of incorporating learned system components, especially those that are shared between mutually-distrusting users. We discuss mitigations and call to action on explicitly addressing known risks when developing learned systems, as well as on studying key types of attacks and defenses to understand their real-world security implications.

## Impact Statement

As ML becomes a core part of database systems, its integration raises new security and privacy concerns—especially when these systems handle sensitive data or are shared by untrusted users. This work exposes a novel attack surface introduced by learned components and provides a framework for analyzing such vulnerabilities. In doing so, it enables more secure design of future systems and guides research on defenses tailored to this emerging threat model. While any disclosure of vulnerabilities carries some risk of misuse, we believe that transparent analysis is critical to building trustworthy systems and advancing the security of learning-based infrastructure.

**Acknowledgments**

We would like to acknowledge our sponsors, who support our research with financial and in-kind contributions: CIFAR through the Canada CIFAR AI Chair program, DARPA through the GARD program, Intel, NFRF through an Exploration grant, NSERC through the Discovery Grant and COHESA Strategic Alliance, Deutsche Forschungsgemeinschaft (DFG, German Research Foundation) through the project ALISON (492020528), and the European Research Council (ERC) through the consolidator grant MALFOY (101043410). Resources used in preparing this research were provided, in part, by the Province of Ontario, the Government of Canada through CIFAR, and companies sponsoring the Vector Institute. We would like to thank members of the CleverHans Lab for their feedback.

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

# A    Poisoning attack on ALEX

## A.1    Attack Construction

Our attack guesses a random position $x_0^0$ in the key range and start inserting keys $x_1^0, x_2^0, ...$ to the left of this position. The following argument proves informally that this results in exponential memory blowup: let $y^0$ be the largest key in the index that is smaller than $x_0^0$. Assume they are in different leaf nodes such that $\alpha \in (y_0, x_0^0)$ where $\alpha$ is the border between $y^0$ and $x^0$ (i.e., smaller keys than $\alpha$ are mapped to $y_0$'s node, and those larger to $x_0^0$), and let $A_{x^0}$ be the leaf node that $x_0^0$ was inserted into. Since subtree ranges are static, as long as the attacker's keys $\{x_n^0\}$ are all larger than $\alpha$, each new insertion will hit the leftmost position within the range that was allocated to $A_{x^0}$ (even if $A_{x^0}$ has split many times by then). If $x_0^0$ and $y_0$ are in the same leaf node initially, then due to their large gap and ALEX's policy of splitting nodes in the middle of their key range, there is likely an insertion $x_0^i$ that causes a split such that $x_0^i$ and $y_0$ are no longer in the same node, let $A_{x^0}$ be that node and the same argument follows from there.

At some point, depending on the sparsity of the indexed keys, the attacker may insert a key that is smaller than $\alpha$. To avoid this, our attacker simply chooses a new random location $x_0^1$ after every $k$ insertions, causing a new parent node size to grow exponentially. Our attacker's locations may hit two child nodes of the same parent node, i.e., $i < j, parent(A_{x^i}) = parent(A_{x^j})$, in which case $A_{x^j}$ already has many redundant pointers (because its parent already doubled its size multiple times). Initially, there would be no need to double the size of the parent array, but soon enough, the exponentially-consumed pointers will be depleted and — since the parent array is likely already at its maximal size — a downwards split will be performed, after which a new parent node starts growing exponentially (that is, at least until insertion $n$ where $x_n < \alpha$).

## A.2    Attack Evaluation

One notable difference between the two datasets considered in the evaluation, is that it takes more insertions and consequently more time to exhaust 512GB memory for the Longitudes dataset (350 million vs 280 million).

To verify if this is due to its relatively higher key density, we check how many of the attacker's keys, on average across the attacker iterations $\{i\}$, fall closer to $y^i$ than to $x_0^i$. For the YCSB dataset, this was only about 1400 on average, whereas Longtitudes, around 5000 out of 10000 keys on average fall closer to $y^i$. Recall that once the keys cross some $\alpha \in (x_0^i, y_0^i)$, the attack ceases to be effective until the next iteration, and this likely happens more frequently for the Longtitudes dataset.

