# OpenReview forum: "Learned-Database Systems Security"
_TMLR — Accepted by TMLR_

### Review · Reviewer_hUsv · 2025-04-15

**Summary Of Contributions:**

This work studies scenarios where a learned model is a sub-component of a system, thus the attacker only has access to the overall system behavior/IO rather than the model's IO. It first presents a overview and meta-analysis about the new attack vectors in this scenario and then present three case studies. First, they present how a side-channel attack (monitoring runtime) can leak important information. Second, they present how a poisoning attack on a learned indexing system leads to memory overflows. Third, they present how to extract information from a worst-case-guaranteed indexing system.

This work presents important progress on the study of learned models embedded in a system.

**Audience:**

Yes

**Broader Impact Concerns:**

No.

**Claims And Evidence:**

No

**Requested Changes:**

I sincerely hope the authors can complete their study on the stated title, "Learned Systems Security," and then submit it to a top venue. This is an important topic. Otherwise, I would suggest the authors change the title and restructure their storyline to present the case studies consistently.

In its current form, it is hard to make minimal requests. This manuscript reads quite unsuitable for publication for the authors' interest, although I will not present a strong objection if the authors insist and restructure their storyline.

**Strengths And Weaknesses:**

Strength:
1. The topic is really important
2. The studies are novel and reveal important insights


Weakness:

The main weakness is that this work feels incomplete, since it consists of a meta-analysis and several case studies. It feels that either the authors should provide a more holistic analysis rather than isolated case studies and submit to a top venue, e.g., SoK track of IEEE S&P, or re-write their story to highlight the case studies in a consistent way. In its current form, it feels like a good paper in the half way and good studies presented in a bad way.

---

> ### Author Response · Authors · 2025-05-28
>
> Dear Reviewer,
>
> Thank you for your constructive feedback! We appreciate your comments on the structure and presentation. We agree that explicitly centering the narrative around security implications of ML components within database systems would improve both the clarity and coherence of our investigation. We revised the manuscript and title accordingly.
>
> Please find the revised version of the paper attached to our OpenReview submission. Major changes are highlighted. In addition, we made many smaller edits throughout the manuscript to ensure consistency and streamlined Section 3 / Table 1 to improve the overall focus on learned database components. We'd be happy to hear any further thoughts you may have.
>
> Best,
> The Authors

---

> > ### Comment · Reviewer_hUsv · 2025-06-03
> >
> > Dear authors,
> >
> > After checking the revision, I still feel the same presentation problem persists. However, if the other reviewers are inclined to accept, I will not veto it.

---

### Review · Reviewer_NFgB · 2025-05-18

**Summary Of Contributions:**

summary:

This paper studies the security vulnerabilities introduced by incorporating ML models as internal components of traditional DB systems — so-called learned systems. The authors propose a principled framework for reasoning about vulnerabilities in these systems. They conceptually discuss the framework against a variety of ML-enhanced systems and demonstrate concrete attacks on three: (1) BAO query optimizer (information leakage), (2) ALEX index (availability attack), and (3) PGM index (privacy leakage via timing).

**Audience:**

Yes

**Claims And Evidence:**

Yes

**Requested Changes:**

**Weaknesses and suggestions**:

W1. The title can be learned DB systems security for clarity. Also, the first lines of the abstract should mention the subject is DB systems.

W2. Attacks are diverse enough (adversarial, poison, and MIA) but the learned components are not state of the art. Adding a more recent learned system such as Wang, Jiayi, et al. "Cardinality estimation using normalizing flow." The VLDB Journal 33.2 (2024): 323-348. can make the paper more current.

W3. Section 2 differences between learned systems and other ML is not crisp enough. For example, approximate query processing engines are widely considered as learned systems. In an AQP, the learned component actually determines the output quality.

W4. Page 5 typo: “except the that tampering/side-channel” -> the

W5. Before section 2, it is necessary to formalize and introduce some background: a) defining learned system as this can be subjective b) giving a small background to different attacks. This can even be done in the introduction.

W6. Having a diagram summarizing the framework and its steps would be nice.

W7. Can section 2.2 be introduced as defend mechanisms?

**Strengths And Weaknesses:**

**Strengths**:

S1. The authors address an important and timely topic that has remained a gap in the literature, i.e. the lack of systematic analysis of the security implications of ML integration in DB systems.

S2. The paper is clearly written, with helpful figures, and adequate reasoning.

S3. The paper goes beyond theory and actually demonstrates practical attacks on real learned systems (BAO, ALEX, PGM). These are well-known learned systems, increasing the credibility and of the results.

S4. I believe this study can have a positive impact by opening venue for future works whereby when ML is incorporated into dB systems, the privacy/security aspect of it is not neglected.

---

> ### Author Response · Authors · 2025-05-28
>
> Dear Reviewer,
>
> Thank you for your constructive feedback!
>
> >  W1. The title can be learned DB systems security for clarity. Also, the first lines of the abstract should mention the subject is DB systems.
>
> This is a valuable point. We have revised the title and updated the paper to more clearly emphasize that our work focuses on the security implications of ML components within database systems.
>
> > W2. Attacks are diverse enough (adversarial, poison, and MIA) but the learned components are not state of the art. Adding a more recent learned system such as Wang, Jiayi, et al. "Cardinality estimation using normalizing flow." The VLDB Journal 33.2 (2024): 323-348. can make the paper more current.
>
> Thank you for this relevant reference. We added this work into our survey on learned components (Section 3.1 and Table 1).
>
> > W3. Section 2 differences between learned systems and other ML is not crisp enough. For example, approximate query processing engines are widely considered as learned systems. In an AQP, the learned component actually determines the output quality.
> W5. Before section 2, it is necessary to formalize and introduce some background: a) defining learned system as this can be subjective b) giving a small background to different attacks. This can even be done in the introduction.
>
> Thank you for raising this important distinction. While AQP systems indeed use ML, our focus is on learned components embedded in core database infrastructure—such as learned indexes—where ML is used to enhance performance but does not directly determine the output of the system. We have revised Section 2 to more clearly define what we mean by “learned systems”. Additionally, we added additional background on the relevant attacks to the introduction.
>
> > W6. Having a diagram summarizing the framework and its steps would be nice.
>
> We appreciate your suggestion. The steps of the framework are outlined in Section 2.1 and summarized in Figure 3.
>
> > W7. Can section 2.2 be introduced as defend mechanisms?
>
> Section 2.2 is intended to explain why traditional adversarial ML threat models fall short when applied to learned components embedded in real-world systems. In these settings, attackers typically have significantly less visibility into the model and limited control over its inputs and training data. These observations could also inform mitigation strategies, e.g., by further constraining the adversaries capabilities and knowledge.
>
> Please find the revised version of the paper attached to our OpenReview submission. Major changes are highlighted. In addition, we made many smaller edits throughout the manuscript to ensure consistency and streamlined Section 3 / Table 1 to improve the overall focus on learned database components. We'd be happy to hear any further thoughts you may have.
>
> Best,
> The Authors

---

> > ### Comment · Reviewer_NFgB · 2025-06-03
> >
> > Thanks for addressing my comments. I think the paper is in a better shape now.

---

### Review · Reviewer_qAaM · 2025-05-19

**Summary Of Contributions:**

This paper considers adversarial attacks on systems where machine learning has been used to optimize performance of the system, called "learned systems". By using machine learning these systems are more efficient in practice but are also vulnerable to adversarial attacks on the learned part of the system. The authors present a framework to organize the types of possible attacks on any given learned system based on the preconditions of the learned system and whether or not resources are shared. The authors present three case studies showing how such adversarial attacks can be performed on real learned systems. Finally, the authors briefly discuss potential mitigations to such attacks.

**Audience:**

Yes

**Broader Impact Concerns:**

There is no Broader Impact Statement in the paper, but there probably should be, especially since this paper is dealing with security and privacy. The Broader Impact Statement should discuss the potential positive and negative side-effects of this work and how it enables future work on this important area of research.

**Claims And Evidence:**

Yes

**Requested Changes:**

#### High-level (not required for acceptance)
* This paper seems to be written for researchers more familiar with systems and less familiar with ML. I think that especially since this is being published in TMLR an effort should be made to make this more accessible to ML researchers. For example, Figures 1 and 2 could be examples of one of the three real systems used in the paper. Adding more background would be beneficial in making this work more accessible to the TMLR community.
* Explain the practicality of the presented attacks. It is unclear what level of resources an attack would need and what the trade-offs would be for each type of attack.
* The motivation and framework are much broader than the empirical results. Maybe the framework should be qualified around a more specific set of systems and vulnerabilities.

#### Minor
* The caption for Figure 4 should be improved to enable the figure to be interpreted independently from the text.
* There should be x-axis labels on Figure 5
* Table 2 is not easily understood. What are the precision groups, average recall, and % of queries. This table should be more self-contained.

**Strengths And Weaknesses:**

#### Strengths
* The problem of learned system security is a difficult and important problem that deserves more attention.
* The poisoning attack on ALEX is extremely impressive and concerning.
* The paper is very well-written and thorough.

#### Weaknesses
* Besides the ALEX attack, it's unclear the magnitude of impact one of these attacks would have on the system. Are these failures catastrophic?
* "information flow" a critical component of the proposed framework is undefined, but used heavily throughout the paper.
* It looks like most of the example learned systems have shared resources which is what enables the attack. Why does this paper even discuss non-shared resource systems if there are no empirical results and many fewer of the possible attacks apply.
* All of the empirical results are performed on learned indexing systems. Is this the motivating example learned system? Why not run on other learned systems such as schedulers or forecasters?

---

> ### Author Response · Authors · 2025-05-28
>
> Dear Reviewer,
>
> Thank you for your constructive feedback!
>
> > This paper seems to be written for researchers more familiar with systems and less familiar with ML. I think that especially since this is being published in TMLR an effort should be made to make this more accessible to ML researchers. For example, Figures 1 and 2 could be examples of one of the three real systems used in the paper. Adding more background would be beneficial in making this work more accessible to the TMLR community.
>
> We appreciate your observation and agree that additional context would enhance the paper’s accessibility. We updated the manuscript to better define key concepts, particularly “information flow” (Section 2.2) and the types of “learned systems” (Section 2) we focus on. We also improved Figure 4, Figure 5, and Table 2, as suggested.
>
> > Explain the practicality of the presented attacks. It is unclear what level of resources an attack would need and what the trade-offs would be for each type of attack.
>
> We appreciate your suggestion. Our primary objective is to demonstrate the feasibility of the proposed attacks through empirical evaluation of the relevant ML components. For each attack, we detail the experimental setup and outline the assumptions required to make the attack practically realizable under plausible conditions.
>
> > The motivation and framework are much broader than the empirical results. Maybe the framework should be qualified around a more specific set of systems and vulnerabilities.
>
> We agree, and have revised the introduction and description of the framework to more precisely reflect our focus on learned components within database systems (Section 1 and 2). We also updated the title.
>
> > There is no Broader Impact Statement in the paper, but there probably should be, especially since this paper is dealing with security and privacy. The Broader Impact Statement should discuss the potential positive and negative side-effects of this work and how it enables future work on this important area of research.
>
> This is a valuable point. We added a broader impact statement to discuss both the risks and the long-term potential of our research (Section 10).
>
> Please find the revised version of the paper attached to our OpenReview submission. Major changes are highlighted. In addition, we made many smaller edits throughout the manuscript to ensure consistency and streamlined Section 3 / Table 1 to improve the overall focus on learned database components. We'd be happy to hear any further thoughts you may have.
>
> Best,
> The Authors

---

> > ### Comment · Reviewer_qAaM · 2025-06-03
> >
> > Thank you for your response and revisions to the paper. All of my comments and concerns have been addressed. The paper is much improved.

---

### Decision · Action_Editor_tvg8 · 2025-06-19

**Recommendation:** Accept as is

**Audience:**

Yes

**Audience Explanation:**

This manuscript studies new attack surfaces that do not exist in traditional database systems — those without ML components — which may be of interest to the TMLR community for understanding the security implications introduced by integrating ML into database systems or beyond.

**Claims And Evidence:**

Yes

**Claims Explanation:**

The initial version of this manuscript made broader claims—specifically about "learned systems security" — that extended beyond what the empirical evidence could support. However, following discussions with the reviewers, the manuscript now focuses more narrowly on "learned database systems."